# UGT1A1 variants in Chinese Uighur and Han newborns and its correlation with neonatal hyperbilirubinemia

Hui Yang[1☉]*, Huijun Li[2☉], Qingyao Xia[3], Wencheng Dai[2], Xin Li[1], Yan Liu[1], Jie Nie[1], Fei Yang[1], Yunfeng Sun[4], Lei Feng[5], Liye Yang[6]

1 Department of Laboratory Medicine, School of Medicine, Yangtze University, Jingzhou, China, 2 Department of Gynecology and Obstetrics, Maternity and Child Health Care Hospital, Urumqi Municipality Xinjiang Uighur Autonomous Region, Xinjiang Province, China, 3 Department of Laboratory Medicine, Western China Women and Child's Hospital, Sichuan University, Chengdu, China, 4 Department of Rehabilitation, Affiliated Traditional Chinese Medicine Hospital of Xinjiang Medical University, Urumqi Municipality Xinjiang Uighur Autonomous Region, Xinjiang Province, China, 5 Department of Laboratory Medicine, People's Hospital of Yuxi City, Yuxi, P. R. China, 6 Lab for Respiratory Disease, People's Hospital of Yangjiang, Yangjiang, P. R. China

☉ These authors contributed equally to this work.
* yanghui2016@yangtze.edu.cn

**Data Availability Statement:** All relevant data are within the paper and its Supporting Information files.

**Funding:** This study was funded by Natural Science Foundation of China (81801509). The

## Abstract

To explore the correlation between UGT1A1 variant and neonatal hyperbilirubinemia in Chinese Uighur and Han populations. We conducted this study in Urumqi, China. Umbilical cord blood specimens and clinical information of term infants born in the studied center were collected. Variation status of UGT1A1 was determined by direct sequencing or capillary electrophoresis analysis. 102 Uighur and 99 Han normal term neonates, together with 19 hospitalized term newborns (10 Uighur and 9 Han) due to significant hyperbilirubinemia were enrolled into the final analysis. The incidence of neonates with high-risk transcutaneous bilirubin level (TCB) were much higher in Han newborns than in Uighur newborns($P = 0.01$). Also, there was statistically significant difference in (TA) 7 promoter mutation of UGT1A1 between Han and Uighur group($\chi 2 = 4.675$, $P = 0.03$). Furthermore, exon mutation (c.211 and /or c.1091) in UGT1A1 gene was significantly associated with increased TCB level ($OR_{adj} = 1.41$, 95%CI: 0.25–2.51, P = 0.002) and higher risk of hyperbilirubinemia in both Han and Uighur infants after adjusted for covariates ($OR_{adj} = 2.21$, 95%CI: 1.09–4.49, $P = 0.03$). In conclusion, UGT1A1 promoter polymorphism seem to be an important genetic modulator of plasma bilirubin level and neonatal hyperbilirubinemia risk within ethnic groups. Genetic assessment of UGT1A1 coding variants may be useful for clinical diagnosis of neonatal jaundice.

## Introduction

Neonatal jaundice or hyperbilirubinemia frequently manifests as a pediatric complex trait or disorder, which is still prevalent in the newborn population today [1]. Although most of them

funders had no role in study design, data collection and analysis, decision to publish, or preparation of the manuscript.

**Competing interests:** The authors declare that they have no conflict of interest.

were generally benign, a select number of infants will develop hazardous levels of total serum bilirubin (TSB) that may cause irreversible neurological damage [2–4].

The incidence and severity of neonatal hyperbilirubinemia in Asians and American Indians are much higher, as compared to those in Caucasian and black populations [5, 6]. Both environmental and genetic factors may contribute to this situation. The importance of genetic role in neonatal hyperbilirubinemia has been increasingly recognized [7].

Innate ethnic variation in UGT1A1, which encoded for the key enzyme involved in the conjugation of bilirubin has been reported to contribute to the biologic basis of hyperbilirubinemia risk in Asian. Several polymorphisms in the promoter and coding region of the UGT1A1 gene complex has been described, which was associated with reduction in UGT1A1 enzyme activity and Gilbert syndrome phenotype-an autosomal recessive unconjugated hyperbilirubinemia (UCH) disorder [8–10]. Of these UGT1A1 variants, the coding sequence variants-UGT1A1*6 polymorphism was predominant in East Asian subjects [11, 12]. And UGT1A1 promoter sequence polymorphisms, the most common in the Caucasian population and frequently associated with GS phenotype, were also observed in East Asian subjects, though at lower allele frequencies [13].

China was famous for its large population and was composed of 56 ethnic groups. Recent multi-center epidemiology survey from China revealed that neonatal hyperbilirubinemia was still common in China (34.4% of term neonates in China). Hazardous hyperbilirubinemia remained not rare, especially in some areas of southern China [14]. Our previous studies had confirmed the strong association between the common coding variant G211A (UGT1A1*6) and severe hyperbilirubinemia in Han population of southern China [15–17].

In this study, we aim to further describe the spectrum and prevalence of UGT1A1 variant in newborns of Uighur minority living in northern China. Uighur was one main minority in northern China, with populations of more than 5 million. Growing genetic studies has demonstrated that Uighur living in Xinjiang Uighur Autonomous Region had quite different genetic background which were quite distant from the subpopulations of Chinese Han, Hui and Mongol populations [18, 19]. The study of genetic differentiation across populations would shed insight into the genetic basis of hyperbilirubinemia risk in Asian newborns.

## Methods

### Study design and sample collection

This study was conducted in the obstetric and newborn ward of Maternal and Child Care Center in Urumqi, Xinjiang Uighur Autonomous Region, China. The study was approved by Ethics Committee of Yangtze University and the hospital.

After informed consent was obtained from the maternal, umbilical cord blood specimens were collected from the newborns born in the obstetric center of Urumqi from March 2018 to December in 2019 by the obstetrician. Clinical records including the birth date, ethnic group, birth weight, delivery method, gestational age, the subsequent transcutaneous bilirubin (TCB) levels in forehead and chest within 1–7 hospital days were collected, Apar score were reviewed. Eligible neonates were those single vaginal delivered term neonates with a gestational age of more than 37 weeks, a birth weight >2500 g and no major birth abnormalities and serious illness, Apar = 10. The ethnic of the newborns was inferred from the self-reported demographic information (name and race) of their parents by reviewing the clinical record. If both their parents were Uighur or Han newborns were included in this study.

Additionally, peripheral blood samples were collected from newborns with continuous hyperbilirubinemia in the pediatric center of the studied hospital between March to September in 2018. Samples were collected and stored at -20˚C after completion of the routine blood test

and the data were analyzed anonymously. The ethical aboard approved a waiver of written consent.

Hyperbilirubinemia was diagnosed and treated according to the updated clinical guidelines of the Chinese Medical Association for neonates [20]. The recorded peak TCB and/or TSB was used to divide the study subjects into case and control subjects. The case subjects included jaundiced neonates with a maximum TSB /TCB that reached or above P95 percentile value of the hour-specific TSB/TCB nomogram drawn by Chinese Multicenter Study Coordination Group for Neonatal Hyperbilirubinemia [21].

### Molecular analysis

The genomic DNA was extracted from surplus EDTA anti-coagulated cord blood specimens using FlexiGene DNA Kit (Qiagen Inc, Valencia, California). The promoter, all five exons, and exon-intron boundaries, of UGT1A1 were tested by polymerase chain reaction (PCR) amplification with 5 pair of primers and direct sequencing as previously described [15–17]. Repeat polymorphism (TA)n in the promoter region was further confirmed by capillary electrophoresis analysis, detailed in our previous studies [15].

### Data analysis

Hardy-Weinberg equilibrium (HWE) test and Linkage disequilibrium (LD) analysis for the identified variants in the UGT1A1 locus was performed by using the web tool SNPStats (https://snpstats.net/start.htm), as we have done in our previous study [16, 17].

Differences in the categorical variables within the two groups were compared by Chi-square test or Fisher's exact test. Independent group t-test or was used to analyze the difference of continuous variables if the dataset was normally distributed; otherwise, the Mann-Whitney test was used.

A linear regression model was used to assess the association between specific polymorphisms or haplotypes of UGT1A1 and the recorded peak TcB values. Logistic regression models were performed to evaluate the association between the specific UGT1A1 variants and the development of neonatal hyperbilirubinemia categorized by the recorded peak TCB levels. Multiple logistic regression analysis was used to evaluate the independence of genetic variants of UGT1A1 with the neonatal hyperbilirubinemia risk after adjusting for known clinical risk factors for neonatal hyperbilirubinemia, including sex, age and ethnic group. The independent variable has been "case" or "control"; a step-down procedure retaining only those variables with $P$ value $<0.1$ was used. UGT1A1 genotype was forced into the model. Furthermore, the association analysis for each UGT1A1 variant was performed under different genetic model assumptions (co-dominant, dominant or recessive).

All analyses were conducted using SPSS version 16.0 (SPSS Inc., Chicago, IL, USA). $P<$ 0.05 was considered statistically significant.

## Results

### Demographic and clinical characteristics

Finally, a total of 201 term newborns (102 Uighur and 99 Han) from the obstetric ward, together with 19 term newborns (10 Uighur and 9 Han) with significant hyperbilirubinemia (TSB>171umol/L) enrolled in the pediatric center of the studied hospital were taken into the final analysis.

Basic demographic characteristics of 201 term newborns without major birth abnormalities and serious illness were summarized in Table 1. Newborns of Han ethnic group and Uighur ethnic group were analyzed separately. Of the selected factors obtained from chart records,

**Table 1. Demographic and clinical features of neonates in Han and Uighur group.**

|  | Uighur (n = 102) | Han (n = 99) | P |
|---|---|---|---|
| Sex |  |  | NS |
| male | 52(51.0%) | 52(52.5%) |  |
| female | 50(49.0%) | 47(47.5%) |  |
| Gestational age(week) | 39.7±0.11 | 39.6+0.09 | NS |
| Birth weight(kg) | 3.45±0.04 | 3.27±0.05 | NS |
| Maximum TCB(mg/dl) |  |  |  |
| Head TCB | 10.71±0.29 | 11.91±0.27 | 0.003 |
| Chest TCB | 11.03±0.29 | 12.15±0.28 | 0.006 |
| Neonate jaundice |  |  | 0.01 |
| Yes (TCB≥12.9mg/dl) | 25(32.9%) | 39(53.3%) |  |
| No (TCB<12.9mg/dl) | 51 (67.1%) | 34(46.6%) |  |
|  |  |  | - |

NS: No significant

except for the mean TCB levels, the other factors including gestational age, gender, birth weight showed no statistically significant difference between the Han and Uighur newborns. Newborns of Han population has a significantly higher TCB level compared with Uighur newborns. Moreover, the incidence of neonates with high-risk TCB level (according to the hour-specific transcutaneous bilirubin nomogram for neonatal hyperbilirubinemia by the Chinese Multicenter Study Coordination Group) within 1–3 days showed statistically higher in Han normal term newborns (39/73) than in Uighur normal term newborns (25/76) *(P = 0.01)*.

## UGT1A1 variant

In addition to the variation in the promoter UGT1A1*28 [A(TA)6TAA/A(TA)7TAA (6/7), rs81753472] and phenobarbital response enhancer module of UGT1A1(-3275 T>G, rs4124874), two variant sites within the coding region of this gene were identified, includingUGT1A1*6(c.211G>A, p.Arg71Gly, rs4148323), *73(c.1091C>T, p.Pro364Leu, rs34946978) (Fig 1). c.211G>A mutation was the predominant exon mutation observed in the study cohort. Specially, the total exon mutation incidence, homozygous mutation rate and heterozygous mutation rate of UGT1A1 were 2.1% (2/96) and 22.9% (22/96) in Uighur newborns, and 2.3% (2/86) and 29.1% (25/86) in Han newborns. As to the (TA)n polymorphism, four kinds of genotype (TA)5/6, 6/6, 6/7 and 7/7 were found (Fig 2), the frequency of TA6/TA7 were 39.2% (40/102) in Uighur newborns and 23.4% (22/94) in Han newborns. Genotype (TA) 7/7 and (TA) 5/6 were observed in one case each in Uighur newborns.

None of the polymorphisms showed statistically significant deviations from the HWE in the study subjects except T-3279G polymorphism, which was excluded from the sub-analysis. A strong pairwise LD was observed between the two identified coding SNP (|D|' = 0.9566, rs4148323 and rs34946978).

There was no statistical difference in c.211G>A mutation frequency between Han and Uighur group, but there was statistically significant difference of (TA)n polymorphism in the promoter region of UGT1A1 gene between Han and Uighur group (*P* = 0.02) (Table 2).

## UGT1A1 variant and neonatal hyperbilirubinemia

Neonates with homozygous and heterozygous UGT1A1 coding mutation (c.211 G>A and/or c.1091 C>T) showed higher TCB level as compared to those with wild UGT1A1 genotype in

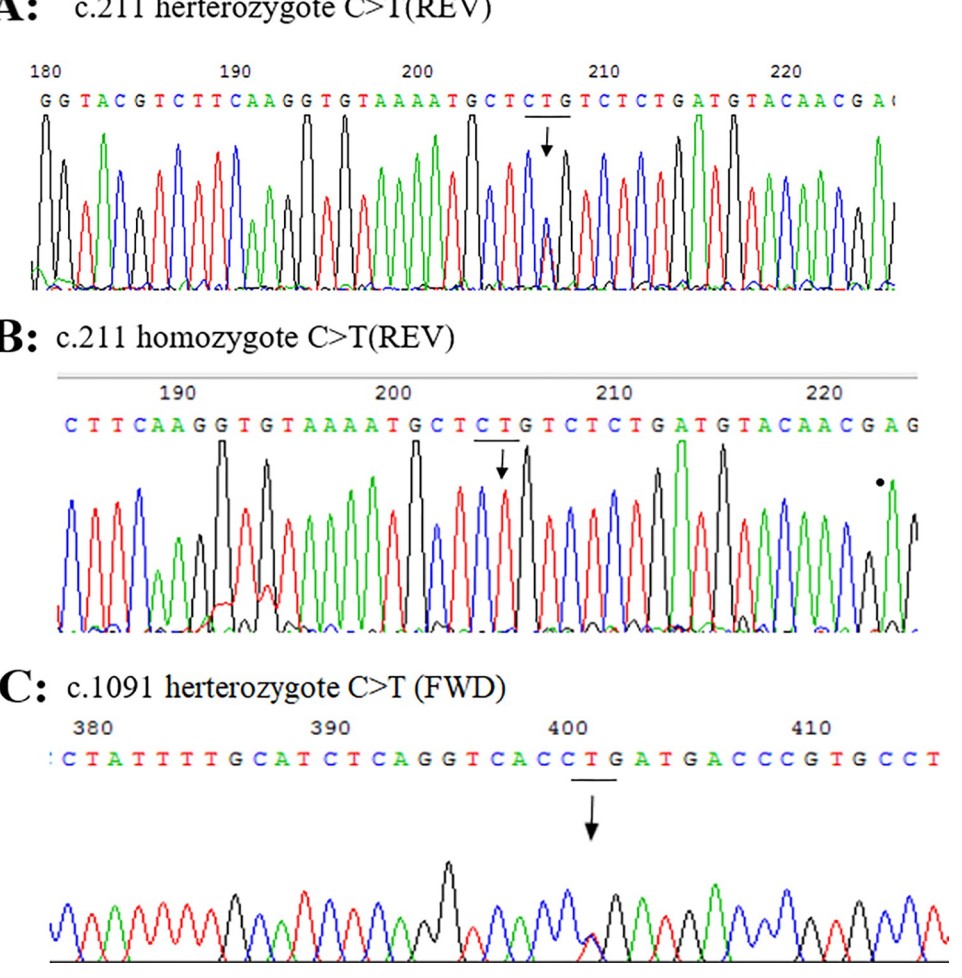

**Fig 1. Mutations of UGT1A1 found in our study cohort.** (A) c. 211 G >A heterozygote (Gly71Arg); (B) c. 211 G >A homozygote (Gly71Arg) (C) c.1091C>T heterozygote (Pro364Leu).

both Han and Uighur neonate after adjusted for the potential covariance. ($OR_{adj}$ = 2.36, 95% CI = 1.14–4.88, $P$ = 0.002), while this trend was not observed for the $(TA)_n$ promoter variant (Table 3). Moreover, when analysis of TCB level in neonates clustered according to the UGT1A1 haplotype, it was showed that homozygous mutation including combined heterozygotes variants in the promoter and the exon region of UGT1A1 significantly increased the TCB level of the studied neonates ($P<0.05$) (Table 4).

In addition, logistic regression indicated that exon mutation (c.211 and /or c.1091) in UGT1A1 gene was associated with increased risk of hyperbilirubinemia in both Han and Uighur neonates. This strong association of UGT1A1 exon mutation with neonatal hyperbilirubinemia remained statistically significantly after adjusting for known clinical risk factors for neonatal hyperbilirubinemia including gender, age and ethnic group. Specifically, neonates who carried heterozygous or homozygous variation in the exon of UGT1A1 had a 2.21(95% CI:1.09–4.49) fold risk of hyperbilirubinemia as compared with those having the wild genotype ($P$ = 0.03). Moreover, the risk of hyperbilirubinemia was even higher in those newborns with a homozygous exon mutation [9.26 (95%CI: 1.77–48.39), P = 0.008]. As for the (TA)7 promoter variant, homozygous (TA)7 also seemingly associated with increased risk of neonatal

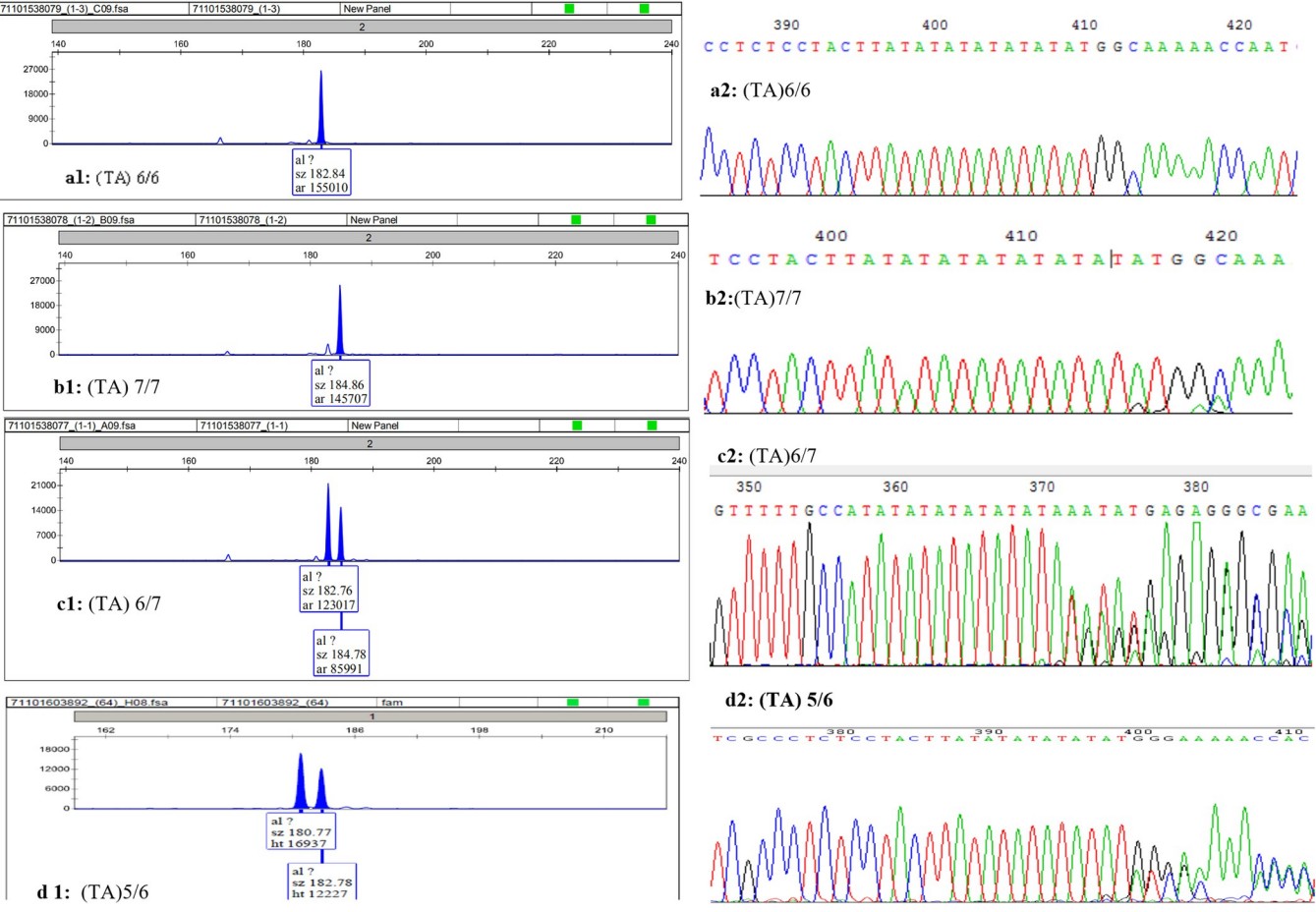

**Fig 2. A typical chromatograph of capillary electrophoresis of UGT1A1 (TA)n promoter polymorphism followed by direct sequencing.** a(1–2) (TA)6 /(TA)6 homozygote; b(1–2) (TA)6/(TA)7 heterozygote. c(1–2) (TA)7 /(TA)7 homozygote; d(1–2) (TA)5 /(TA)6 homozygote.

hyperbilirubinemia (P = 0.05, Table 5). Similarly, haplotype analysis showed that (TA)6-A ((TA)n -rs4148323) significantly increased the risk of neonatal hyperbilirubinemia ($OR_{adj}$ = 2.23; $P$ = 0.015) compared to the most common haplotype (TA)6-G.

## Discussion and conclusion

Asian neonates were regarded as high-risk population for severe hyperbilirubinemia. The genetic background across population was an important risk factor modulating this disorder. UGT1A1, one of main factors for pathologic hyperbilirubinemia in newborns, varied within and between populations [8, 22]. In this study, we found that the incidence of neonatal hyper-bilirubinemia was much higher in full-term Chinese Han newborns than in Uighur newborns of Chinese population. Notably, our data also showed a significant different distribution of promoter variation in UGT1A1- encoding for hepatic bilirubin metabolism enzyme in Han and Uighur. Taken together, we inferred that promoter TATA box variation might be a risk factor that modulated neonatal hyperlipidemia risk in Chinese Han and Uighur newborns. Our results were consistent with Beutler et al. study in African, Europe and Asia population, which has firstly reported the genetic difference of TA7 and serum bilirubin level among populations, they proposed that the (TA)n repeat might be a balanced polymorphism evolutionarily selected to maintain serum bilirubin in an optimal range in the face of largely undefined

**Table 2. Genotypes frequency of c.-3275T>G in enhance, (TA)$_n$ repeat polymorphism and *6 (c.211G > A, p.Arg71Gly) variant of UGT1A1 gene in Han vs Uighur groups.**

| | *Han* | | Uighur | | $P_{allele}$ | $P_{genotype}$ |
|---|---|---|---|---|---|---|
| | n | $P_{H-W}{}^a$ | n | $P_{H-W}{}^a$ | | |
| (TA)$_n$ | | 0.35 | | 0.07 | 0.02 | 0.02 |
| TA$_5$/ TA$_6$ | 0 | | 1 | | | |
| TA$_6$/ TA$_6$ | 72 | | 60 | | | |
| TA$_6$/ TA$_7$ | 22 | | 40 | | | |
| TA$_7$/ TA$_7$ | 0 | | 1 | | | |
| G211A variation | | 1 | | 0.68 | 0.38 | 0.63 |
| G/G | 59 | | 72 | | | |
| G/A | 25 | | 22 | | | |
| A/A | 2 | | 2 | | | |
| c.-3275T>G | | 0.32 | | 0.03 | 0.33 | |
| T/T | 28 | | 28 | | | 0.04 |
| T/G | 29 | | 66 | | | |
| G/G | 13 | | 16 | | | |

a: Hardy-Weinberg Equilibrium test p value.

genetic and environmental pressures [23]. Our study further confirmed this hypothesis. Moreover, considering the difference of TA7 mutation rate and bilirubin level in Asia and Caucasian population, the data obtained in the study added more evidence to previous genetic studies in Uighur ethnic group which has suggested that Uighur ethnic group was a gene admixture of Eastern Asian and European populations, and much closed to European than other ethnics in China [18, 19].

UGT1A1 coding sequence variant c.211 G>A (UGT1A1*6, G71R), common variant predominant in Asia population while not found in Caucasian population, was the main cause of

**Table 3. The associations between TCB level and UGT1A1 mutation adjusted by age, gender and race: Linear regression analysis.**

| | Han | | | Uygur | | | Total | | |
|---|---|---|---|---|---|---|---|---|---|
| | Mode[#] | OR$_{adj}$[†] (95%CI) | P | Mode[#] | OR$_{adj}$[†] (95%CI) | P | Mode[#] | OR$_{adj}$[†] (95%CI) | P |
| TATA box | - | | NA | Recessive | | 0.22 | Recessive | | 0.26 |
| TA6/TA6+TA6/TA7 | | reference | | | reference | | | reference | |
| TA7/TA7 | | - | | | 3.87(-2.21–9.95) | | | 3.85(-2.88–10.57) | |
| c.211 G>A | Dominant | | 0.046 | Recessive | | 0.02 | Dominant | | 0.03 |
| G/G | | reference | | | reference | | | reference | |
| G/A | | 1.78(0.06–3.51) | | | | | | 1.31(0.14–2.49) | |
| A/A | | | | | 3.82(0.07–6.93) | | | | |
| Compound exon mutation | Dominant | | 0.046 | Recessive | | 0.005 | Dominant | | 0.02 |
| Wildtype | | reference | | | reference | | | reference | |
| Heterozygote | | 1.78(0.06–3.51) | | | | | | 1.41(0.25–2.57) | |
| Homozygote[*] | | | | | 3.74(1.19–6.29) | | | | |

[#] In the linear regression analysis, neonates with wild UGT1A1 genotype (i.e. G/G) was set as reference group under the dominant genetic model assumption, while both those wildtype and the heterozygous UGT1A1 variant carrier (i.e. G/G+G/A) were set as the reference group under the recessive model.

NA: not available

†Adjusted for gender, birth week, race.

*Including compound heterozygous c.211 G>A variant plus heterozygous c.1091 C>T variant.

**Table 4. TCB levels of study neonates in the subgroup divided by UGT1A1 genotype.**

| | Han | | Uygur | | Total | |
|---|---|---|---|---|---|---|
| | n | TCB(mg/dl) | n | TCB(mg/dl) | n | TCB(mg/dl) |
| **Haplotype (TA$_n$-rs4148323)** | | | | | | |
| ①66GG | 37 | 12.17±2.85 | 33 | 11.25±3.43 | 70 | 11.74±3.15 |
| ②67GG | 14 | 13.00±5.05 | 26 | 10.56±3.14 | 40 | 11.41±4.03 |
| ③66GA | 23 | 13.32±3.07 | 13 | 10.78±3.68 | 36 | 12.40±3.48 |
| ④67GA | 2 | 16.20±6.81 | 4 | 11.04±2.70 | 6 | 12.88±4.56 |
| ⑤66AA | 4 | | 3 | 14.16±2.55 | 7 | 14.91±5.06 |
| ⑥67AA | 0 | | 2 | | 2 | |
| ⑦77AA | 0 | | 1 | | 1 | |
| **Model 1** | | | | | | |
| ① | 37 | 12.17±2.85 | 33 | 11.25±3.43 | 70 | 11.74±3.15 |
| ②+③ | 37 | 13.20±3.88 | 39 | 10.64±3.28 | 76 | 11.88±3.79 |
| ④+⑤+⑥+⑦ | 6 | 16.20±6.81 | 10 | 12.91±2.94 | 16 | 14.15±4.83 |
| $P_a$ | | 0.047/0.07/0.016 | | 0.155/0.056/0.169 | | 0.05/0.025/0.018 |
| **Model 2** | | | | | | |
| ①+② | 51 | 12.4±3.55 | 59 | 10.95±3.29 | 110 | 11.62±3.48 |
| ③+④ | 25 | 13.58±3.35 | 39 | 10.84±3.40 | 42 | 12.47±3.59 |
| ⑤+⑥+⑦ | 4 | 16.03±8.0 | 6 | 14.16±2.55 | 10 | 14.91±5.06 |
| $P_b$ | | 0.114/0.23/0.067 | | 0.072/0.025/0.036 | | 0.017/0.053/0.007 |

a: Difference comparison for ① vs ②+③ vs ④+⑤+⑥+⑦ and for ②+③ vs ④+⑤+⑥+⑦ and ① vs④+⑤+⑥+⑦ within the 3 group.

b: Difference comparison for ①+② vs ③+④ vs ⑤+⑥+⑦ and for ③+④ vs ⑤+⑥+⑦ and ①+② vs⑤+⑥+⑦ within the 3 group.

GS in Asia population [11, 12]. In this study, we confirmed the strong association of the UGT1A1 coding sequence variant c.211 G>A (UGT1A1*6, G71R) with neonatal hyperbilirubinemia risk in both Han and Uighur newborns in China. Moreover, our data showed no statistic differences in the frequency of c.211G>A variant among Chinese Han and Uighur newborns. Therefore, on the one hand, the results here demonstrated that Chinese ethic groups shared a clade, though high genetic differentiation exists among Uighur minority and other Chinese ethnic groups. At the same time, it further demonstrated the necessity of screening of this exon mutation to identify high-risk neonates who tend to be suffered from severe neonatal hyperbilirubinemia in all Chinese ethnic groups. On the other hand, our results revealed that c.211 variant could not explain the difference of neonatal hyperbilirubinemia risk in Chinese Han and Uighur newborns, though its predominant role in Chinese neonatal hyperbilirubinemia risk was certain.

As described previously, the (TA)n repeat variant in the UGT1A1 promoter was another extensively studied variant. As for the influence of genetic factors on the UGT1A1 enzyme activity, previous studies believed that the transcription level of UGT1A1 was mainly affected by TATA polymorphisms in the UGT1A1 promoter [23]. Promoters containing seven thymine adenine(ta) repeats have been found to be less active than the wild-type six repeats, and the serum bilirubin levels of persons homozygous or even heterozygous for seven repeats have been found to be higher than those with the wild-type six repeats [24–26]. However, the role of this variant on neonatal hyperbilirubinemia risk was not yet defined. i.e. A(TA)7TAA variations (UGTA*28), was described as the common cause of Gilbert's syndrome in the Caucasian population, whereas most previous studies in east Asian countries failed to find this association

**Table 5. Associations of UGT1A1 variants and neonate hyperbilirubinemia under different inheritage model assumptions: Logistic regression analysis.**

| | N(%) | | Model 1# | | | Model 2# | | |
|---|---|---|---|---|---|---|---|---|
| | Case | Control | Model | $OR_{adj}$ (95% CI)† | P | Model | $OR_{adj}$ (95% CI)† | P |
| (TA)$_n$ | | | Codominant | | 0.1 | Recessive | | 0.05 |
| TA6/TA6 | 55(0.71) | 69(0.15) | | 1 | | | 1 | |
| TA6/TA7 | 20(0.26) | 36(0.34) | | 0.73(0.36–1.49) | | | | |
| TA7/TA7 | 2(0.03) | 0(0.00) | | 0 (0.00-NA) | | | 0 (0.00-NA) | |
| c.211G>A | | | Codominant | | 0.02 | Dominant | | 0.03 |
| G/G | 44(0.60) | 67(0.74) | | 1 | | | 1 | |
| G/A | 22(0.30) | 22(0.24) | | 1.70(0.79–3.63) | | | 2.15(1.05–4.88) | |
| A/A | 7(0.10) | 2(0.02) | | 7.98(1.46–43.56) | | | | |
| Compound exon mutation | | | Codominant | | 0.008 | Dominant | | 0.03 |
| Wildtype | 43(0.59) | 67(0.74) | | 1 | | | 1 | |
| Heterozygote | 21(0.29) | 22(0.24) | | 1.61(0.75–3.48) | | | 2.21(1.09–4.49) | |
| Homozygote* | 9(0.12) | 2(0.02) | | 9.26(1.77–48.39) | | | | |
| Haplotype analysis (TAn-rs4148323) | Frequency | | | | | | | |
| TA6-G | 0.60 | 0.69 | | 1 | | | | |
| TA6-A | 0.24 | 0.14 | | 2.01(1.06–3.79) | 0.03 | | | |
| TA7-G | 0.13 | 0.17 | | 0.92(0.44–1.96) | 0.84 | | | |
| TA7-A | 0.03 | 0.00 | | 4.82(0.38–60.76) | 0.22 | | | |

\# (0, 1,2) set on the three genotype (wildtype, heterozygote and homozygote) for each variant under the codominant genetic model assumption by multiple regression analysis; (0, 1) set on the genotype data (eg: GG+G/A vs AA; GG vs G/A+AA) for each variant under recessive or dominant inheritance model by binary test.

†Adjusted for gender, birth week, race.

\* Including compound heterozygous c.211 G>A variant plus heterozygous c.1091 C>T variant.

[27, 28]. Recently, one meta-analysis had identified that homozygous while not heterozygous (TA)7 was associated with increased risk of neonatal hyperbilirubinemia in both Asia and European population [29]. In this study, two case of homozygous (TA)7 was observed in Uighur neonate with significant hyperbilirubinemia. Further large-scale population research was necessary to verify the effect of TATA box on the risk of hyperbilirubinemia in the newborns.

We acknowledged that this study has some limitations. First, this study was limited by the small number of neonates enrolled. Some of the trends observed in the study could have reached statistical significance if the study sample had been larger (P = 0.05, Tables 4 and 5). Second, our inability to assess the serum bilirubin for most of the studied neonates, so we used TCB instead of TSB to subgroup our study cohort. Previous studies have shown a linear relationship exists between TcB and TSB, though the correlation significantly decreased when the TCB was high(>229umol/l) [30]. Recently, a system review and meta-analysis by Yu et al also showed that TCB nomogram was as efficient as TSB nomogram for identifying subsequent significant hyperbilirubinemia [31]. Thus, it's applicable to use TcB nomograms to identify neonatal hyperbilirubinemia in this study.

In conclusion, screening for UGT1A1 coding region 211 G to A variation should be taken into consideration to identify the risk group who tend to be suffered from severe neonatal hyperbilirubinemia in all Chinese populations. What's more, our results supported previous hypothesis that the importance of UGT1A1 promoter polymorphism in regulation of plasma bilirubin level within ethnic groups. In future studies, it should be necessary to increase the number ethnic groups analyses in order to assess more data about the genetic basis of neonatal hyperbilirubinemia risk from the main ethnic groups of the Chinese population.

## Supporting information

**S1 Checklist.**
(DOCX)

**S1 File.**
(PDF)

**S2 File.**
(XLSX)

## Author Contributions

**Conceptualization:** Hui Yang.

**Data curation:** Hui Yang, Huijun Li, Qingyao Xia, Wencheng Dai, Yan Liu, Jie Nie, Liye Yang.

**Formal analysis:** Hui Yang.

**Funding acquisition:** Hui Yang.

**Investigation:** Huijun Li, Qingyao Xia, Wencheng Dai, Xin Li, Lei Feng.

**Methodology:** Hui Yang, Huijun Li, Qingyao Xia, Wencheng Dai, Xin Li, Yan Liu, Jie Nie, Fei Yang, Yunfeng Sun.

**Project administration:** Hui Yang.

**Resources:** Hui Yang, Huijun Li, Wencheng Dai, Jie Nie, Fei Yang, Yunfeng Sun, Lei Feng.

**Supervision:** Hui Yang.

**Validation:** Hui Yang.

**Writing – original draft:** Hui Yang, Yan Liu.

**Writing – review & editing:** Hui Yang, Huijun Li, Qingyao Xia, Wencheng Dai, Liye Yang.

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
