## [Decision Letter · Decision Letter 0]

5 Jan 2022

PONE-D-21-33656UGT1A1 variants in Chinese Uighur and Han newborns and its correlation with neonatal hyperbilirubinemiaPLOS ONE

Dear Dr. Yang,

Thank you for submitting your manuscript to PLOS ONE. After careful consideration, we feel that it has merit but does not fully meet PLOS ONE’s publication criteria as it currently stands. Therefore, we invite you to submit a revised version of the manuscript that addresses the points raised during the review process.

Please revise this manuscript completely according to the reviewer's comments and then resubmit.

We look forward to receiving your revised manuscript.

Kind regards,

Mingqing Xu

Academic Editor

PLOS ONE

Journal Requirements:

This study was funded by Natural Science Foundation of China (81801509).

Additional Editor Comments:

Please revise this manuscript completely according to the reviewer's comments and then resubmit.

Reviewers' comments:

Reviewer's Responses to Questions

**Comments to the Author**

1. Is the manuscript technically sound, and do the data support the conclusions?

Reviewer #1: No

Reviewer #2: No

2. Has the statistical analysis been performed appropriately and rigorously? 

Reviewer #1: No

Reviewer #2: No

3. Have the authors made all data underlying the findings in their manuscript fully available?

Reviewer #1: Yes

Reviewer #2: No

4. Is the manuscript presented in an intelligible fashion and written in standard English?

Reviewer #1: Yes

Reviewer #2: No

5. Review Comments to the Author

Reviewer #1: In “UGT1A1 variants in Chinese Uighur and Han newborns and its correlation with neonatal hyperbilirubinemia”, the authors aim to identify genetic variants that are associated with neonatal hyperbilirubinemia in Chinese Uighur and Han populations. Given a higher prevalence of hazardous hyperbilirubinemia in some populations, it is imperative to understand the underlying genetic, and thus, biological mechanisms of this condition. By analyzing populations with varying ancestral backgrounds, the authors aim to identify genetic variants that are associated with hyperbilirubinemia.

There are several major limitations to the current manuscript. One, the datasets are very small and without general population-level information (common population allele frequencies) about the genetic variants , it is difficult to ascertain the strength of the associations presented. Secondly, greater clarity is needed in regards to the statistical analysis methods and presentation of results. As they are currently formatted, the tables (e.g., Tables 3 and 4) are confusing. In their current state it is unclear how many tests were computed (important for multiple comparison correction). It is also currently unclear why some rows appear to be missing p-values or data (more detail, below). If there were count thresholds (e.g. allele counts) that were used to exclude certain comparisons (e.g., in instances of complete separation between cases and controls), then that should be explicitly explained in the methods and table captions.

Details----------

Introduction:

Need an reference for introduction statement “quite distant from the subpopulations of Chinese Han, Hui, and Mongol populations”. Is this based in genetic admixture and ancestral analyses?

Methods:

Given the usage of chi-square or Fisher’s exact test, need to clarify that whether all participants were unrelated, as this is an important methodological consideration. Statistical assumptions of independence would be invalidated if participants were genetically related (e.g., inclusion of siblings or cousins).

Data Analysis: How were the two common variants in the UGT1A1 locus identified? In Reference 15, there are 4 variants within the UGT1A1 locus. Was a specific MAF required? Was this only coding variants?

It would be best to use rsIDs when possible, throughout the paper. This enables readers to more easily connect the presented results to past publications. For instance, it appears that the Arg71Gly corresponds to rs4148323, based on dbSNP and Table 4. However, I’m unable to identifiy the rsID for Pro364Leu, can this be clarified?

Is there a difference between using TCB or TSB for diagnosis of hyperbilirubinemia? Is there a potential source of variation (unintended bias) introduced by using different methods for diagnosis? Can the authors show that there was equal (not-skewed) usage fo TSB/TCB across groups? The usage of two methods is mentioned in the discussion (page 13) but the implication is not explained.

Can the authors clarify how genotype was coded within the logistic regression models? Was this an additive model (e.g., coded 0,1,2) or was it a binary test (0,1) for each genotype (e.g., GG vs G/A+AA; G/A vs GG+AA). In Table 4, it is unclear why some cells appear inconsistently incomplete -please explicitly specify (e.g. in the table caption) which tests were run.

Table 3 appears incomplete. Please better clarify why p-values are presented for some comparisons, but not others. Also, total counts for Compound Exon Mutation are not filled-in for Controls. What does the p=0.015 refer to under Heterozygote for compound exon mutation? Based on the data, it appears the comparison (and statistical test) should be between 20 cases and 20 controls (which presumably does not yield a p-value of 0.015). Please ensure the appropriate data is presented.

Furthermore, the authors should better clarify the exact comparisons for each p-value. What does the p-value of 0.046 assess (under TAn/ 7/7)? From the table, it appears that it compares the 7/7 repeat between cases and controls, but this would not be a statistically feasible comparison with complete separation of counts (0 7/7 copies for controls).

Results:

Given the proximity of the tested variants, what is the linkage disequilibrium among the coding SNPs?

Could the authors include a graph that shows the TSB/TCB levels by genotype and case/control status? It would be helpful to see the effect, respective to the outcome.

Figures:

In figure 1, there is a Typo in parts A and C for heterozygote.

Reviewer #2: In the manuscript entitled “T UGT1A1 variants in Chinese Uighur and Han newborns and its correlation with neonatal hyperbilirubinemia”, the authors conducted an association test by use of 102 Uighur and 99 Han normal term neonates. The novelty is limited and the scientific questions are not addressed deeply.

(1) I suggest the authors conduct a meta-analysis based on the selected SNPs. to see if the findings are specific to a give ethnic or to all populations, the following papers can be cited and followed for the meta-analytic procedures (if the data is not enough available, at least DISCUSSION should be added as the LIMITATION of this study with enough citation to support the viewpoints): Ref 1: Wu Y, et al. Multi-trait analysis for genome-wide association study of five psychiatric disorders. Transl Psychiatry. 2020 Jun 30;10(1):209. Ref 2: Jiang L, et al. Sex-Specific Association of Circulating Ferritin Level and Risk of Type 2 Diabetes: A Dose-Response Meta-Analysis of Prospective Studies. J Clin Endocrinol Metab. 2019 Oct 1;104(10):4539-4551. Ref 3: Wang X, Wu W, Zheng W, Fang X, Chen L, Rink L, Min J, Wang F. Zinc supplementation improves glycemic control for diabetes prevention and management: a systematic review and meta-analysis of randomized controlled trials. Am J Clin Nutr. 2019 Jul 1;110(1):76-90.

(2) The interactions between environmental and biological factors should also be explored to see specific SNPs are respond to the environmental factors, especially I suggest using mendelian randomization analysis to test if UGT1A1 promoter polymorphism causally trigger the risk of neonatal hyperbilirubinemia through mediating the plasma bilirubin level. If cannot, please discuss the limitations in the Discussion in detail with additional citations to support the viewpoints. For these reasons, the following papers regarding causal inference between genetic variants, inter-mediator phenotype and disease outcome can be cited and followed.

Reference 1: Fuquan Zhang, Ancha Baranova, Chao Zhou, et al. Causal influences of neuroticism on mental health and cardiovascular disease. Human Genetics. 2021 May 1

Reference 2:Fuquan Zhang, et al. Genetic evidence suggests posttraumatic stress disorder as a subtype of major depressive disorder. Journal of Clinical Investigation. 2021 May 30

Reference 3:Xinhui Wang, et al. Genetic support of a causal relationship between iron status and type 2 diabetes: a Mendelian randomization study. The Journal of Clinical Endocrinology & Metabolism. 2021 June 19

(3) In addition, the significantly associated SNPs may be used to predict disease susceptibility, therefore, the authors may explore the possibility to conduct a machine-learning model to predict disease risk based these significant SNPs. For this reason, the authors may cite the following papers to follow these references’ procedure to construct a standard prediction model based on the significant SNPs (probably include the environmental factors). Especially deep learning method is a very promising way to predict disease risk based on clinical information and genetic biomarkers (If deep learning can not be used, please discuss as the LIMITATION of this study with enough citation to support the viewpoints). Ref 4: Yu H, et al. LEPR hypomethylation is significantly associated with gastric cancer in males. Exp Mol Pathol. 2020 Oct;116:104493. Ref 5: Liu M, et al. A multi-model deep convolutional neural network for automatic hippocampus segmentation and classification in Alzheimer's disease. Neuroimage. 2020 Mar;208:116459.

6. PLOS authors have the option to publish the peer review history of their article (what does this mean?). If published, this will include your full peer review and any attached files.

Reviewer #1: No

Reviewer #2: No

---

## [Author Response · Author response to Decision Letter 0]

19 Feb 2022

Journal Requirements:

Answer: Thank you for your suggestion. We have prepared and upload the manuscript as suggested and carefully proof-read the manuscript to minimize the mistakes and typographic, grammatic errors.

This study was funded by Natural Science Foundation of China (81801509).

Answer: The statement for funder

 ("The funders had no role in study design, data collection and analysis, decision to publish, or preparation of the manuscript.") is corrected and not need to amend.

Answer: Thank you for your suggestion. We have removed the phrase that refers to these data as suggested in the revised manuscript.

4,5. Please include captions for your Supporting Information files at the end of your manuscript, and update any in-text citations to match accordingly.

Answer: Thank you for your suggestion. We have revised and upload the manuscript as suggested and carefully proof-read the manuscript to minimize the mistakes and typographic, grammatic errors.

#Responds to Review 1

Reviewer #1: In “UGT1A1 variants in Chinese Uighur and Han newborns and its correlation with neonatal hyperbilirubinemia”, the authors aim to identify genetic variants that are associated with neonatal hyperbilirubinemia in Chinese Uighur and Han populations. Given a higher prevalence of hazardous hyperbilirubinemia in some populations, it is imperative to understand the underlying genetic, and thus, biological mechanisms of this condition. By analyzing populations with varying ancestral backgrounds, the authors aim to identify genetic variants that are associated with hyperbilirubinemia.

There are several major limitations to the current manuscript. 

One, the datasets are very small and without general population-level information (common population allele frequencies) about the genetic variants , it is difficult to ascertain the strength of the associations presented. 

Secondly, greater clarity is needed in regards to the statistical analysis methods and presentation of results. As they are currently formatted, the tables (e.g., Tables 3 and 4) are confusing. In their current state it is unclear how many tests were computed (important for multiple comparison correction). It is also currently unclear why some rows appear to be missing p-values or data (more detail, below). If there were count thresholds (e.g. allele counts) that were used to exclude certain comparisons (e.g., in instances of complete separation between cases and controls), then that should be explicitly explained in the methods and table captions.

Answer: Thank you for your suggestion. As the review point that the study is limited by the small size of the study cohort. Our current study is an observation study. However, the outbreak of COVID-19 makes the sample collection really hard. As we described in the method, all the studied sample were collected from 2018.3-2019.12, and stopped when the outbreak of covid-19 in Wuhan in the initial of 2020. Finally, a total of 102 Uighur and 99 Han normal term neonates. We fully acknowledge this limitation. Further large-scale population study is need to identify the association of UGT1A1 mutation and neonatal hyperbilirubinemia in multiple ethnic groups of the Chinese population. We have presented this limitation in the discussion part of the revised manuscript. Also, there are some errors in the table 3 and table 4. We have revised these errors as suggested and marked red in the new version. 

Details----------

Introduction:

Need an reference for introduction statement “quite distant from the subpopulations of Chinese Han, Hui, and Mongol populations”. Is this based in genetic admixture and ancestral analyses?

Answer: Thank you for your suggestion. Two references [18, 19] about the genetic admixture analysis of Chinese Uighur and other Chinese population have been cited in the revised version.

[18] Suhua Zhang, Yingnan Bian, Li Li, Kuan Sun, Zheng wang, Qi Zhao, et al. Population genetic study of 34 X-Chromosome markers in 5 main ethnic groups of China. Sci Rep 2015; 5: 17711. 

[19] Zhang Z, Wei S, Gui H, Yuan Z, Li, S. The contribution of genetic diversity to subdivide populations living in the silk road of China. PLoS One 2014; 9, e97344

Methods:

Given the usage of chi-square or Fisher’s exact test, need to clarify that whether all participants were unrelated, as this is an important methodological consideration. Statistical assumptions of independence would be invalidated if participants were genetically related (e.g., inclusion of siblings or cousins).

Answer: Thank you for your suggestion. All the studied participants (the pregnancy woman and newborn) were recruit from one single hospital from March 2018 to December 2019. And twin gestation was excluded. It’s certain that there are no siblings in our studied neonates. However, whether cousins were included in our studied neonates that was unable to judge according to the information we obtained.

Data Analysis: How were the two common variants in the UGT1A1 locus identified? In Reference 15, there are 4 variants within the UGT1A1 locus. Was a specific MAF required? Was this only coding variants?

Answer: Thank you for your suggestion. As we described in the manuscript, the promoter, all five exons, and exon-intron boundaries, and a region in the distal promoter (the Phenobarbital response enhancer module) of UGT1A1 were tested by polymerase chain reaction (PCR) amplification with 5 pair of primers and direct sequencing. All the UGT1A1 variants including the two common variants-(TA)n repeat variant(dbSNP rs81753472) and Arg71Gly(rs4148323) were identified by direct sequencing. (TA)n repeat variant was further confirmed by capillary electrophoresis analysis.

In Reference 15, there are 4 variants within the UGT1A1 locus. Was a specific MAF required? Was this only coding variants?

Answer: Thank you for your suggestion. The kind and MAF of UGT1A1 variants were varied in different population. As you see, 4 kinds of coding variants were found in reference 15, while, only 2 kinds of UGT1A1 coding variants (Arg71Gly and Pro364Leu) were identified in this study population.

It would be best to use rsIDs when possible, throughout the paper. This enables readers to more easily connect the presented results to past publications. For instance, it appears that the Arg71Gly corresponds to rs4148323, based on dbSNP and Table 4. However, I’m unable to identifiy the rsID for Pro364Leu（rs34946978）, (c.1091C>T, p.Pro364Leu), can this be clarified?

Answer: Thank you for your suggestion, rsID of the identified UGT1A1 variants were added in the revised manuscript.

Is there a difference between using TCB or TSB for diagnosis of hyperbilirubinemia? Is there a potential source of variation (unintended bias) introduced by using different methods for diagnosis? Can the authors show that there was equal (not-skewed) usage fo TSB/TCB across groups? The usage of two methods is mentioned in the discussion (page 13) but the implication is not explained.

Answer: Previous studies have shown a linear relationship exists between TcB and TSB, though the correlation significantly decreased when the TCB was high(>229umol/l) [30]. Recently, a system review and meta-analysis by Yu et al also showed that TCB nomogram was as efficient as TSB nomogram for identifying subsequent significant hyperbilirubinemia. The pooled diagnosis AUC of TCB nomogram was 0.817 which showed no significantly different to the summary TSB nomogram (0.819) [31]. Thus, it’s applicable to use TcB nomograms to identify neonatal hyperbilirubinemia in this study. A system review study by Yu et al. have showed that TcB nomograms had the same predictive value as TSB nomograms for diagnosis of hyperbilirubinemia. we have added explanations for it in the revised version.

[30] Frias C,Kolman KB,Mathieson KM.A comparison of transcutaneous and total serum bilirubin in newborn Hispanic infants at 35 or more weeks of gestation.[J].Journal of the American Board of Family Medicine: JABFM,2007,20(3).

[31] Zhang-Bin Yu, Shu-Ping Han, Chao Chen. Bilirubin nomograms for identification of neonatal hyperbilirubinemia in healthy term and late-preterm infants: a systematic review and meta-analysis. World J Pediatr. 2014,(3).211-218.

Can the authors clarify how genotype was coded within the logistic regression models? Was this an additive model (e.g., coded 0,1,2) or was it a binary test (0,1) for each genotype (e.g., GG vs G/A+AA; G/A vs GG+AA). In Table 4, it is unclear why some cells appear inconsistently incomplete -please explicitly specify (e.g. in the table caption) which tests were run.

Table 3 appears incomplete. Please better clarify why p-values are presented for some comparisons, but not others. Also, total counts for Compound Exon Mutation are not filled-in for Controls. What does the p=0.015 refer to under Heterozygote for compound exon mutation? Based on the data, it appears the comparison (and statistical test) should be between 20 cases and 20 controls (which presumably does not yield a p-value of 0.015). Please ensure the appropriate data is presented.

Furthermore, the authors should better clarify the exact comparisons for each p-value. What does the p-value of 0.046 assess (under TAn/ 7/7)? From the table, it appears that it compares the 7/7 repeat between cases and controls, but this would not be a statistically feasible comparison with complete separation of counts (0 7/7 copies for controls).

Answer: Thank you for your question. As the reviewer pointed that the results presented in Tables (table 3 and 4) is confusing, we have amended and modified the presentation of these results in the new tables (table 3 and table 5). 

Results:

Given the proximity of the tested variants, what is the linkage disequilibrium among the coding SNPs?

Answer: Thank you for your suggestion. Linkage disequilibrium analysis was done for the coding SNPs. A strong pairwise LD was observed between the two identified coding SNP (|D|’=0.9566, rs4148323 and rs34946978). We have described this in the revised manuscript and marked red.

Could the authors include a graph that shows the TSB/TCB levels by genotype and case/control status? It would be helpful to see the effect, respective to the outcome.

Answer: Thank you for your suggestion. Tables (Table 3 and 4) have added for showing the TCB levels by genotype and case/control status in the new version.

Figures:

In figure 1, there is a Typo in parts A and C for heterozygote.

Answer: Thank you for your suggestion. we have amended the mistake in the new version.

Responds to Review 2

Reviewer 2: In the manuscript entitled “T UGT1A1 variants in Chinese Uighur and Han newborns and its correlation with neonatal hyperbilirubinemia”, the authors conducted an association test by use of 102 Uighur and 99 Han normal term neonates. The novelty is limited and the scientific questions are not addressed deeply.

Answer: Thank you for your suggestion. Increasing population study has explore the association between UGT1A1 variant and neonatal hyperbilirubinemia，and confirm the important role of UGT1A1 c.211 variant in Asian neonatal hyperbilirubinemia. However, as to the role of TA promoter in serum bilirubin level and neonatal hyperbilirubinemia risk, the result is controversial. Eg: Meta -analysis study conducted in 2011 doesn’t found its association with neonatal hyperbilirubinemia in Asia, though its significant associated with neonatal hyperbilirubinemia in white population [28]. Recently, another meta-analysis have found that TA promoter was significantly associated in both Asia and causian population. The precise role of TA promoter in serum bilirubin level and neonatal hyperbilirubinemia risk merit further study. That’s the reason why we conduct this study in two main different ethic population in China. Our current study is an observation study. However, the outbreak of COVID-19 makes the sample collection really hard. As we described in the method, all the studied sample were collected from 2018.3-2019.12, and stopped when the outbreak of covid-19 in Wuhan in the initial of 2020. Finally, a total of 102 Uighur and 99 Han normal term neonates. We fully acknowledge that the present study is limited by the small study sample sized. We have presented this limit in the discussion part of the revised manuscript.

(1) I suggest the authors conduct a meta-analysis based on the selected SNPs. to see if the findings are specific to a give ethnic or to all populations, the following papers can be cited and followed for the meta-analytic procedures (if the data is not enough available, at least DISCUSSION should be added as the LIMITATION of this study with enough citation to support the viewpoints): Ref 1: Wu Y, et al. Multi-trait analysis for genome-wide association study of five psychiatric disorders. Transl Psychiatry. 2020 Jun 30;10(1):209. Ref 2: Jiang L, et al. Sex-Specific Association of Circulating Ferritin Level and Risk of Type 2 Diabetes: A Dose-Response Meta-Analysis of Prospective Studies. J Clin Endocrinol Metab. 2019 Oct 1;104(10):4539-4551. Ref 3: Wang X, Wu W, Zheng W, Fang X, Chen L, Rink L, Min J, Wang F. Zinc supplementation improves glycemic control for diabetes prevention and management: a systematic review and meta-analysis of randomized controlled trials. Am J Clin Nutr. 2019 Jul 1;110(1):76-90.

Answer: Thank you for your suggestion. As the reviewer point that our current study is an observation study which is limited by the small-scale study sample. As we know, there are two papers by Chinese researchers has reported the meta-analysis study on UGT1A1 variant and neonatal hyperbilirubinemia [28, 29], however, the result is somewhat inconsistent. As to the role of TA promoter in serum bilirubin level and neonatal hyperbilirubinemia risk, the meta -analysis study conducted in 2011 doesn’t found its association with neonatal hyperbilirubinemia in Asia, though its significant associated with neonatal hyperbilirubinemia in white population [28]. That’s the reason why we conduct this study in two main different ethic population in China. Recently, another meta-analysis study (2020) has found that TA promoter was significantly associated in both Asia and European population. We have cited these two meta-analysis studies and added the discussion in my revised manuscript.

[28]Long J, Zhang S, Fang X, Luo Y, Liu J. Association of neonatal hyperbilirubinemia with uridine diphosphate-glucuronosyltransferase 1A1 gene polymorphisms: Meta-analysis. Pediatrics International 2011; 53:530-540.

[29]Jing Wang, Jiansong Yin, Mei Xue, Jun Lyu, Yu Wan. Roles of UGT1A1 Gly71Arg and TATA promoter polymorphisms in neonatal hyperbilirubinemia: A meta-analysis. Gene. 2020;736:144409.

(2) The interactions between environmental and biological factors should also be explored to see specific SNPs are respond to the environmental factors, especially I suggest using mendelian randomization analysis to test if UGT1A1 promoter polymorphism causally trigger the risk of neonatal hyperbilirubinemia through mediating the plasma bilirubin level. If cannot, please discuss the limitations in the Discussion in detail with additional citations to support the viewpoints. For these reasons, the following papers regarding causal inference between genetic variants, inter-mediator phenotype and disease outcome can be cited and followed.

Reference 1: Fuquan Zhang, Ancha Baranova, Chao Zhou, et al. Causal influences of neuroticism on mental health and cardiovascular disease. Human Genetics. 2021 May 1

Reference 2:Fuquan Zhang, et al. Genetic evidence suggests posttraumatic stress disorder as a subtype of major depressive disorder. Journal of Clinical Investigation. 2021 May 30

Reference 3:Xinhui Wang, et al. Genetic support of a causal relationship between iron status and type 2 diabetes: a Mendelian randomization study. The Journal of Clinical Endocrinology & Metabolism. 2021 June 19

Answer: Thank you for your suggestion. As the reviewer point that both environment and genetic factors contributed to neonatal hyperbilirubinemia. In this study, all our studied neonates (both Han and Uyghur) all from the same area, and in the case-control association analysis, we have adjusted the potential risk factor like birth week, gender and race. We have detailed this in the method part of this manuscript.

(3) In addition, the significantly associated SNPs may be used to predict disease susceptibility, therefore, the authors may explore the possibility to conduct a machine-learning model to predict disease risk based these significant SNPs. For this reason, the authors may cite the following papers to follow these references’ procedure to construct a standard prediction model based on the significant SNPs (probably include the environmental factors). Especially deep learning method is a very promising way to predict disease risk based on clinical information and genetic biomarkers (If deep learning can not be used, please discuss as the LIMITATION of this study with enough citation to support the viewpoints). Ref 4: Yu H, et al. LEPR hypomethylation is significantly associated with gastric cancer in males. Exp Mol Pathol. 2020 Oct;116:104493. Ref 5: Liu M, et al. A multi-model deep convolutional neural network for automatic hippocampus segmentation and classification in Alzheimer's disease. Neuroimage. 2020 Mar;208:116459.

Answer: Thank you for your suggestion. Machine-learning model is a very promising way to predict disease risk based on clinical information and genetic biomarkers. We have fully known that our present study is limited by the small size of the study cohort. we would learn to use machine-learning model to further explore the association of ugt1a1 gene mutation and neonatal hyperbilirubinemia in the further research.

---

## [Decision Letter · Decision Letter 1]

18 Jul 2022

PONE-D-21-33656R1UGT1A1 variants in Chinese Uighur and Han newborns and its correlation with neonatal hyperbilirubinemiaPLOS ONE

Dear Dr. Yang,

Thank you for submitting your manuscript to PLOS ONE. After careful consideration, we feel that it has merit but does not fully meet PLOS ONE’s publication criteria as it currently stands. Therefore, we invite you to submit a revised version of the manuscript that addresses the points raised during the review process. Your manuscript has been assessed by the two reviewers who reviewed the original submission. Reviewer has provided some comments; please respond to these in your revised manuscript. Please also respond to the requests I have added below my signature.

We look forward to receiving your revised manuscript.

Kind regards,

George Vousden

Deputy Editor in Chief

PLOS ONE

Additional Editor Comments:

A) Please outline how ethnic group was determined in your Methods section. Since any method of identifying an ethnic group suffers from limitations, please discuss the limitations of the method(s) used for this manuscript in your Discussion section.

Reviewers' comments:

Reviewer's Responses to Questions

**Comments to the Author**

1. If the authors have adequately addressed your comments raised in a previous round of review and you feel that this manuscript is now acceptable for publication, you may indicate that here to bypass the “Comments to the Author” section, enter your conflict of interest statement in the “Confidential to Editor” section, and submit your "Accept" recommendation.

Reviewer #1: (No Response)

Reviewer #2: All comments have been addressed

2. Is the manuscript technically sound, and do the data support the conclusions?

Reviewer #1: Yes

Reviewer #2: Yes

3. Has the statistical analysis been performed appropriately and rigorously? 

Reviewer #1: Yes

Reviewer #2: Yes

4. Have the authors made all data underlying the findings in their manuscript fully available?

Reviewer #1: Yes

Reviewer #2: Yes

5. Is the manuscript presented in an intelligible fashion and written in standard English?

Reviewer #1: Yes

Reviewer #2: Yes

6. Review Comments to the Author

Reviewer #1: The authors have worked to address previous concerns. The addition of SNP rsIDs, additional references, and clarifications about the differences between TSB and TCB make the manuscript much clearer. The authors have also better clarified the limitations of the study given the sample size. So while these small numbers provide limits on the types of analyses possible, they do present the relative allele counts which could help inform focus of future studies in larger cohorts.

Only remaining comment is on Table 3. It's unclear why some rows have p-values but not odds ratios (and vice versa). Also, the inclusion of 0.00 odds ratios is confusing, is this meant to indicate there were equal numbers of cases/controls with that particular genotype? It might be clearer to list the counts, and then the p-value next to the Odds ratio. If I'm misunderstanding, perhaps horizontal lines would help clarify how the data is meant to align.

Reviewer #2: The authors have improved their manuscript as suggested, questions are all well addressed. I have not additional concerns.

7. PLOS authors have the option to publish the peer review history of their article (what does this mean?). If published, this will include your full peer review and any attached files.

Reviewer #1: No

Reviewer #2: No

---

## [Author Response · Author response to Decision Letter 1]

19 Aug 2022

Additional Editor Comments:

Please outline how ethnic group was determined in your Methods section. Since any method of identifying an ethnic group suffers from limitations, please discuss the limitations of the method(s) used for this manuscript in your Discussion section.

Answer：Thank you for your suggestion. We have clarified this question in the revised manuscript. In this study, blood samples (umbilical cord and peripheral blood species) of healthy unrelated newborns from the two main ethnic groups (Uygur and HAN) were collected. The ethnic composition of the newborns was determined according to the ethnic identity of his father and mother. Both their parents were Uygur or Han were included in this study. In fact, intermarriage between Uyghur and Han is rare, though it is allowed by the law of China. 

Reviewer #1: Only remaining comment is on Table 3. It's unclear why some rows have p-values but not odds ratios (and vice versa). Also, the inclusion of 0.00 odds ratios is confusing, is this meant to indicate there were equal numbers of cases/controls with that particular genotype? It might be clearer to list the counts, and then the p-value next to the Odds ratio. If I'm misunderstanding, perhaps horizontal lines would help clarify how the data is meant to align.

Answer: Thank you for your question. We have adjusted table 3 and make it easier to see as suggested.

---

## [Decision Letter · Decision Letter 2]

21 Oct 2022

PONE-D-21-33656R2UGT1A1 variants in Chinese Uighur and Han newborns and its correlation with neonatal hyperbilirubinemiaPLOS ONE

Dear Dr. Yang,

Thank you for submitting your manuscript to PLOS ONE. After careful consideration, we feel that it has merit but does not fully meet PLOS ONE’s publication criteria as it currently stands. The concerns previously noted by the reviewer have been addressed. However, my previous concerns about how ethnicity was determined have not been addressed satisfactorily. It is indicated that "the ethnic composition of the newborns was determined according to the ethnic identity of his father and mother", but how the ethnic composition was determined remains unclear. Please update your Methods section to provide enough details of how ethnicity was determined, such that another researcher could repeat the procedures with the information provided. In your Discussion section, please discuss whether it was possible that the ethnicity of any of the participants was incorrect. 

We look forward to receiving your revised manuscript.

Kind regards,

George Vousden

Deputy Editor in Chief

PLOS ONE

Journal Requirements:

Reviewers' comments:

Reviewer's Responses to Questions

**Comments to the Author**

1. If the authors have adequately addressed your comments raised in a previous round of review and you feel that this manuscript is now acceptable for publication, you may indicate that here to bypass the “Comments to the Author” section, enter your conflict of interest statement in the “Confidential to Editor” section, and submit your "Accept" recommendation.

Reviewer #1: All comments have been addressed

2. Is the manuscript technically sound, and do the data support the conclusions?

Reviewer #1: Yes

3. Has the statistical analysis been performed appropriately and rigorously? 

Reviewer #1: Yes

4. Have the authors made all data underlying the findings in their manuscript fully available?

Reviewer #1: (No Response)

5. Is the manuscript presented in an intelligible fashion and written in standard English?

Reviewer #1: Yes

6. Review Comments to the Author

Reviewer #1: Yes, in revisiting Table 3, it is now clearer why certain genotypes (e.g., those that were the reference genotypes) did not have test results.

The samples sizes are small, but the authors highlight this in limitations and list the counts.

7. PLOS authors have the option to publish the peer review history of their article (what does this mean?). If published, this will include your full peer review and any attached files.

Reviewer #1: No

---

## [Author Response · Author response to Decision Letter 2]

24 Oct 2022

Response to the academic editor and reviewers

1. The concerns previously noted by the reviewer have been addressed. However, my previous concerns about how ethnicity was determined have not been addressed satisfactorily. It is indicated that "the ethnic composition of the newborns was determined according to the ethnic identity of his father and mother", but how the ethnic composition was determined remains unclear. Please update your Methods section to provide enough details of how ethnicity was determined, such that another researcher could repeat the procedures with the information provided. In your Discussion section, please discuss whether it was possible that the ethnicity of any of the participants was incorrect

Answer：Thank you for your suggestion. We have clarified this question more precisely in the revised manuscript. The ethnic of the newborns was inferred from the demographic information (name and race) of their parents by reviewing the clinical record. Both their parents were Uygur or Han were included in this study. In fact, there are so many differences between Uygur and Han, such as their appearance, costume, linguistic, naming way and so on, so that it was not difficult to differential one from the other. Moreover, the demographic information of the study participants was carefully check not less than three times by us, it was not possible that mistaken the ethnicity of any of the participants.

Journal Requirements:

Answer：No change.

---

## [Editor Report · Decision Letter 3]

8 Nov 2022

PONE-D-21-33656R3UGT1A1 variants in Chinese Uighur and Han newborns and its correlation with neonatal hyperbilirubinemiaPLOS ONE

Dear Dr. Yang,

Thank you for submitting your manuscript to PLOS ONE. After careful consideration, we feel that it has merit but does not fully meet PLOS ONE’s publication criteria as it currently stands. Therefore, we invite you to submit a revised version of the manuscript that addresses the points raised during the review process.

We look forward to receiving your revised manuscript.

Kind regards,

George Vousden

Deputy Editor in Chief

PLOS ONE

Journal Requirements:

Editor Comments:

Thank you for revising your manuscript in response to my previous comments. Please revise the manuscript to respond to the following minor concerns. Once these concerns are addressed the manuscript will be ready for publication:

1) It is noted that "The ethnic of the newborns was inferred from the demographic information (name and race) of their parents by reviewing the clinical record.". Please indicate how race/ethnicity was originally determined when clinical data was collected - please indicate whether ethnicity was self-reported by patients, or whether this was inferred by clinicians.

2) Suggest to revise the sentence "Both their parents were Uygur or Han were included in this study." to " If both their parents were Uygur or Han newborns were included in this study."

3) Please remove this sentence "In fact, there are so many differences between Uygur and Han, such as their appearance, costume, linguistic, naming way and so on, so that it was not difficult to differential one from the other"

4) Please ensure the spelling of Uyghur is consistent throughout the manuscript, as several spellings are currently used (e.g. Uighur in the title and Uygur in the abstract and other sections of the manuscript)

---

## [Author Response · Author response to Decision Letter 3]

9 Nov 2022

Response to the editor 

1) It is noted that "The ethnic of the newborns was inferred from the demographic information (name and race) of their parents by reviewing the clinical record.". Please indicate how race/ethnicity was originally determined when clinical data was collected - please indicate whether ethnicity was self-reported by patients, or whether this was inferred by clinicians.

Answer：Thank you for your suggestion. Demographic information including the race/ ethnicity was self-reported by the patients on their admission. We have detail this in the revised manuscript and marked red.

2) Suggest to revise the sentence "Both their parents were Uygur or Han were included in this study." to " If both their parents were Uygur or Han newborns were included in this study."

Answer：Thank you for your suggestion. I have done the modification as you suggested in the revised manuscript.

3) Please remove this sentence "In fact, there are so many differences between Uygur and Han, such as their appearance, costume, linguistic, naming way and so on, so that it was not difficult to differential one from the other"

Answer：Thank you for your suggestion. We have done the revision as suggested.

4) Please ensure the spelling of Uyghur is consistent throughout the manuscript, as several spellings are currently used (e.g. Uighur in the title and Uygur in the abstract and other sections of the manuscript)

Answer：Thank you for your suggestion. I have done the modification as suggested in the revised manuscript.

---

## [Editor Report · Decision Letter 4]

1 Dec 2022

UGT1A1 variants in Chinese Uighur and Han newborns and its correlation with neonatal hyperbilirubinemia

PONE-D-21-33656R4

Dear Dr. Yang,

We’re pleased to inform you that your manuscript has been judged scientifically suitable for publication and will be formally accepted for publication once it meets all outstanding technical requirements.

Kind regards,

George Vousden

Deputy Editor in Chief

PLOS ONE

Additional Editor Comments:

1) A word is missing from the revised manuscript - I suggested that the sentence "If both their parents were Uygur or Han newborns were included in this study.". However, the word "newborns" is missing from the manuscript. Please revise. 

2) We note that you have indicated that “All relevant data are within the manuscript and its Supporting Information files”. However, members of the editorial team have assessed the provided datafiles and are concerned that the data provided do not meet our expectations for minimal datasets. PLOS defines the minimal data set to consist of the data required to replicate all study findings reported in the article, as well as related metadata and methods (see (https://journals.plos.org/plosone/s/data-availability). For example, authors should submit the following data:

> The values behind the means, standard deviations and other measures reported;

> The values used to build graphs;

> The points extracted from images for analysis.

Please ensure that you have provided a datafile to meet these requirements with your manuscript or provide relevant accession numbers in your Data Availability Statement.
---

## [Editor Report · Acceptance letter]

6 Dec 2022

PONE-D-21-33656R4 

UGT1A1 variants in Chinese Uighur and Han newborns and its correlation with neonatal hyperbilirubinemia 

Dear Dr. Yang:

I'm pleased to inform you that your manuscript has been deemed suitable for publication in PLOS ONE. Congratulations! Your manuscript is now with our production department. 

Kind regards, 

on behalf of

Dr. George Vousden 

Staff Editor

PLOS ONE